# Optimization of Thermomechanical Processing under Double-Pass Hot Compression Tests of a High Nb and N-Bearing Austenitic Stainless-Steel Biomaterial Using Artificial Neural Networks

Gláucia Adriane de S. Sulzbach [1], Maria Verônica G. Rodrigues [1], Samuel F. Rodrigues [1,2,*], Marcos Natan da S. Lima [3], Rodrigo de C. Paes Loureiro [3], Denis Fabrício S. de Sá [4], Clodualdo Aranas, Jr. [5], Glaucia Maria E. Macedo [6], Fulvio Siciliano [1,7], Hamilton F. Gomes de Abreu [3], Gedeon S. Reis [1] and Eden S. Silva [1,6]

1 Graduate Program in Materials Engineering, Federal Institute of Education, Science and Technology of Maranhão-IFMA, São Luís 65075-441, Maranhão, Brazil
2 Materials Science and Engineering Graduate Program, Federal University of Piaui—UFPI, Teresina 64049-550, Piaui, Brazil
3 Materials Characterization Laboratory (LACAM), Department of Metallurgical and Materials Engineering, Federal University of Ceará, Campus Do Pici, Bloco 729, Fortaleza 60020-181, Ceará, Brazil
4 Electric Engineering Department-Balsas (COELE), Federal University of Maranhão—UFMA, Campus Balsas, MA-140, KM 04, Balsas 65800-000, Maranhão, Brazil
5 Mechanical Engineering, University of New Brunswick, Fredericton, NB E3B 5A3, Canada
6 Federal Institute of Education, Science and Technology of Maranhão-IFMA, Campus Barra do Corda, Barra do Corda 65950-000, Maranhão, Brazil
7 Dynamic Systems Inc., 323 NY 355, Poestenkill, NY 12140, USA
* Correspondence: samuel.filgueiras@ifma.edu.br; Tel.: +55-98-98517-9142

**Abstract:** Physical simulation is a useful tool for examining the events that occur during the multiple stages of thermomechanical processing, since it requires no industrial equipment. Instead, it involves hot deformation testing in the laboratory, similar to industrial-scale processes, such as controlled hot rolling and forging, but under different conditions of friction and heat transfer. Our purpose in this work was to develop an artificial neural network (ANN) to optimize the thermomechanical behavior of stainless-steel biomaterial in a double-pass hot compression test, adapted to the Arrhenius–Avrami constitutive model. The method consists of calculating the static softening fraction ($X_s$) and mean recrystallized grain size ($d_s$), implementing an ANN based on data obtained from hot compression tests, using a vacuum chamber in a DIL 805A/D quenching dilatometer at temperatures of 1000, 1050, 1100 and 1200 °C, in passes ($\varepsilon_1 = \varepsilon_2$) of 0.15 and 0.30, a strain rate of 1.0 s$^{-1}$ and time between passes ($t_p$) of 1, 10, 100, 400, 800 and 1000 s. The constitutive analysis and the experimental and ANN-simulated results were in good agreement, indicating that ASTM F-1586 austenitic stainless steel used as a biomaterial undergoes up to $X_s$ = 40% of softening due solely to static recovery (SRV) in less than 1.0 s interval between passes ($t_p$), followed by metadynamic recrystallization (MDRX) at strains greater than 0.30. At T > 1050 °C, the behavior of the softening curves $X_s$ vs. $t_p$ showed the formation of plateaus for long times between passes ($t_p$), delaying the softening kinetics and modifying the profile of the curves produced by the moderate stacking fault energy, $\gamma_{sfe}$ = 69 mJ/m$^2$ and the strain-induced interaction between recrystallization and precipitation (Z-phase). Thus, the use of this ANN allows one to optimize the ideal thermomechanical parameters for distribution and refinement of grains with better mechanical properties.

**Keywords:** physical simulation; thermomechanical processing; austenitic stainless steel; metadynamic recrystallization; precipitation; artificial neural network (ANN)

## 1. Introduction

The main stainless steels used as biomaterials in Brazil's national health system (SUS) today are ASTM F-138 and ASTM F-1586 steels, which are in high demand for orthopedic implants as alternatives to expensive titanium and cobalt-chromium alloys [1]. However, there are numerous records of allergic and pathological reactions in the human body when these metals interact with body fluids [2,3]. Research on ASTM F-1586 steel [4] has sought to improve its performance, cost and functionality, in both the controlled rolling process and hot forging parameters, in order to increase its mechanical strength, corrosion resistance, biocompatibility and workability. The fundamentals of physical simulation of controlled thermomechanical processing are discussed, based on scientific concepts and on the feasibility of optimizing manufacturing on a laboratory scale [5].

Advanced thermomechanical processing with multiple strains has been an alternative for improving the mechanical properties of steels, controlling metallurgical operations, microstructural changes, and hardening and softening mechanisms (static recovery—SRV, static recrystallization—SRX or metadynamic recrystallization—MDRX) [6]. The inconvenience of replicating a manufacturing process has led to the establishment of physical simulations to evaluate mechanical-microstructural aspects, whereby samples are subjected to thermomechanical cycles similar to industrial ones [7]. The plastic behavior is then parameterized by means of constitutive equations, allowing for the development of homogeneous and refined microstructures.

The need to match physical simulation results with metallurgical parameters made it interesting to model the controlled hot rolling process. Sellars et al. [8] were the first to propose an analytical phenomenological model for predicting stress under multiple strains, analyzing the effect of softening mechanisms. An integral stress response on the $\sigma$ vs. $\varepsilon$ curves caused by competing work hardening and dynamic softening behavior was proposed by Estrin–Mecking [9]. Subsequently, Laasraoui et al. [10] developed constitutive-mathematical models to predict stress and microstructural evolution in hot rolling, associating the effects of work hardening and static softening in Avrami kinetics. Medina et al. [11] presented a physical approach, proposing correlation models of grain boundary mobility with the driving force resulting from the interaction between recrystallization and precipitation.

Artificial neural network (ANN) simulation has been increasingly employed in thermomechanical processing, enabling the estimation of nonlinear parameters in the basic neuron model that reflect the material's behavior under hot deformation, developing new workability routes with essential information from the process through synapses and structural organization [12]. Several promising studies in the literature [13] describe the application of ANN in predicting the hot working behavior of metal alloys. Using fuzzy logic, Li et al. [14] developed an ANN model to evaluate the microstructural evolution of TC6 alloy by estimating the volumetric fraction and grain size during hot forging and reported a good agreement with their experimental results. Feng W. et al. [15] used ANN with a generic algorithm to optimize the thermomechanical processing parameters of AISI 304 steel under continuous isothermal conditions.

Narayana et al. [16] investigated the evolution of the microstructure of a chromium-nickel alloy under hot forging and non-isothermal conditions using ANNs. They stated that the essential feature of the model is the lattice code used in the estimation of steady state volume fraction and grain size, which fits well with their predicted results, indicating the exceptional ability of ANNs to predict these parameters. Thus, the application of ANNs in the simulation of thermomechanical processing of steels is aimed at learning patterns presented in the training phase and making correlations between input (T, $\varepsilon$, Q, $t_p$) and output ($\sigma$, $X_s$, $d_s$) parameters based on the Arrhenius–Avrami method in order to reproduce the process efficiently and accurately under other conditions [17–20].

The purpose of this work was to develop an ANN best suited to data learning in order to optimize the thermomechanical behavior of ASTM F-1586 steel under double-pass hot compression tests, using Avrami formalism, given the paucity of studies about the kinetic



competition between static softening and strain-induced precipitation. The method consists of estimating the softening fraction ($X_s$) and average size of recrystallized grains ($d_s$), based on empirical data, using algorithms that reflect the dynamics of hot processing. The novelty of the developed ANN proposed in this investigation enables one to estimate the level of mechanical strength under different conditions and evaluate the hardening and softening mechanisms of this steel by taking into consideration the intrinsic physical-metallurgy variables of the thermomechanical process.

## 2. Materials and Methods

Table 1 describes the chemical composition of the ASTM F-1586 austenitic stainless steel investigated here. Cylindrical test specimens, with dimensions of length 10 mm and diameter 5.0 mm, were welded at their ends onto thin molybdenum discs (thickness, $\Phi = 0.06$ mm) to reduce friction. The test specimens were first vacuum induction-heated applying a heating rate of 10 °C/s up to a solubilization temperature of 1250 °C, where they were held for 300 s to ensure complete homogenization. After this, they were subjected to double-pass hot compression tests in a DIL 805A/D quenching dilatometer (BÄHR Thermo-analyse GmbH), at 1000, 1050, 1100 and 1200 °C, with the application of strain ($\varepsilon_1 = 0.15$ and $\varepsilon_2 = 0.30$) at a strain rate of 1.0 s$^{-1}$, with time between passes ($t_p$) of 0.1, 1.0, 10, 100, 400, 800 and 1000 s, as illustrated in Figure 1a,b.

**Table 1.** Chemical composition of ASTM F-1586 steel (mass%).

| C | Si | Mn | Ni | Cr | Mo | S | P | N | Cu | Nb | Fe |
|---|---|---|---|---|---|---|---|---|---|---|---|
| 0.035 | 0.37 | 4.04 | 10.6 | 20.3 | 2.47 | 0.001 | 0.022 | 0.36 | 0.06 | 0.29 | bal. |

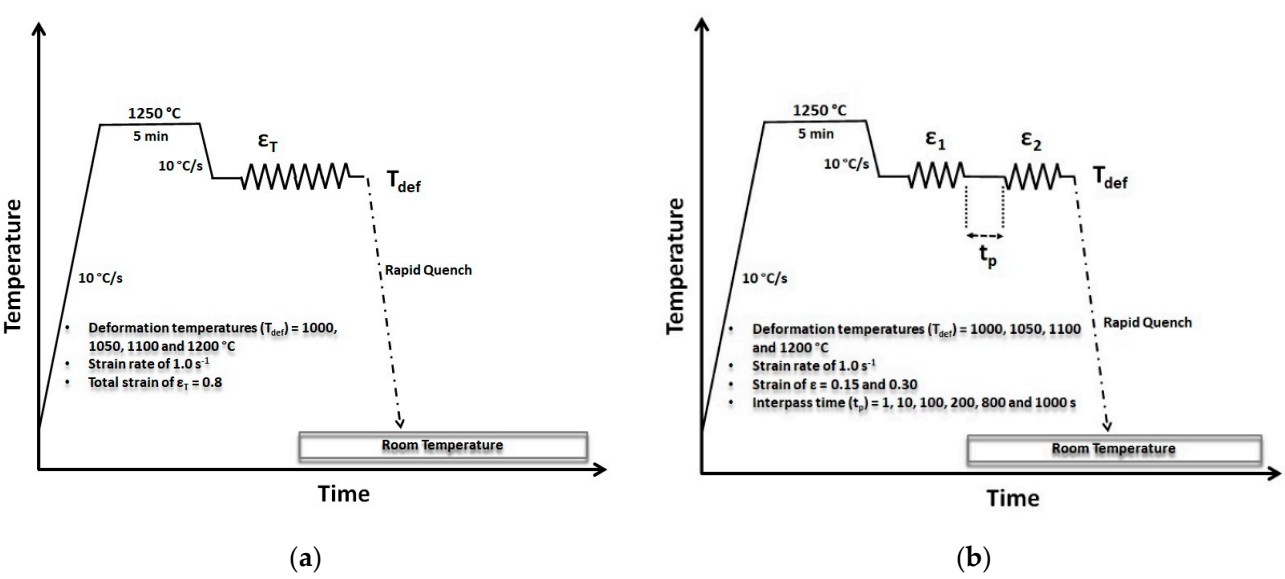

(**a**)  (**b**)

**Figure 1.** Diagram of the thermomechanical cycles applied: (**a**) continuous isothermal compression test; (**b**) double-pass hot compression test.

Based on the experimental stress–strain curves obtained in the double-pass hot compression tests, the softening fraction ($X_s$) was calculated for different temperatures and times between passes ($t_p$) using the 0.2% offset, 2.0% offset and equivalent mean stress (EMS) methods, according to Equations (1) and (2) [17]:

$$X_s = \frac{\sigma_m - \sigma_{y2}}{\sigma_m - \sigma_{y1}} \tag{1}$$

$$\overline{X} = \frac{\overline{\sigma}_m - \overline{\sigma}_y}{\overline{\sigma}_m - \overline{\sigma}_o}, \qquad \overline{\sigma}_y = \frac{1}{\varepsilon_2 - \varepsilon_1} \int_{\varepsilon_1}^{\varepsilon_2} \sigma d\varepsilon, \qquad \overline{\sigma}_o = \frac{1}{\varepsilon_2} \int_0^{\varepsilon_1} \sigma d\varepsilon \tag{2}$$

where $\sigma_m$ is the maximum yield stress of the 1st pass and $\sigma_{y1}$ and $\sigma_{y2}$ are the yield strengths (at 0.2% strain) of the 1st pass and 2nd pass, respectively. In this model, Equation (1) quantifies the effects of static softening. When static recovery or recrystallization does not occur, $\sigma_{y2}$ is equal to $\sigma_m$ and the softening fraction ($X_s$) is equal to zero. On the other hand, when total recrystallization occurs, the softening fraction reaches values close to 100% ($X_s \geq 95\%$). Thus, the softening fraction ($X_s$) varies from 0 to 100%, depending on the evolution of the material's static softening processes and the conditions of hot deformation applied.

The softening resulting from the effects of recovery and recrystallization during unloading between high temperature deformation operations is traditionally quantified in terms of the softening fraction ($X_s$) as a global internal variable, while the softening kinetics is described by the generalized Johnson–Mehl–Avrami–Kolmogorov (JMAK) equation [18] in the form of:

$$X_s = 1 - exp\left[-0.693 \left(\frac{t}{t_{0.5}}\right)^n\right] \tag{3}$$

where $n$ is a material-dependent constant and $t_{0.5}$ is the time elapsed to reach a 50% volume fraction of softening. The constitutive expression of $t_{0.5}$ as a function of the thermomechanical parameters is given by:

$$t_{0.5} = A d_o^r \varepsilon^q Z^p exp\left(\frac{Q}{RT}\right) \tag{4}$$

where $A$, $r$, $q$ and $p$, are constants of the material, $\varepsilon$ is the applied strain, $Z$ is the Zener-Hollomon parameter, $R$ is the gas constant (J/mol·K), $T$ is the absolute temperature (K), $Q$ is the activation strain energy (kJ/mol), and $d_o$ is the initial grain size (µm) [18].

With the softening fraction curves ($X_s$) as a function of time between passes ($t_p$) and the constitutive equations of the softening kinetics, based on Avrami formalism, an artificial neural network (ANN) was developed to optimize the ideal thermomechanical parameters in double-pass hot compression tests. This method is expected to enable investigating and controlling the retardation of static softening kinetics in steel manufacturing processes, ensuring better grain distribution and refinement with improved mechanical properties.

The microstructural evolution of steels during hot forming processes is predicted using physical and phenomenological methods. The behavior of hot deformation of steels is strongly nonlinear, as are many factors that affect the microstructure, which makes the prediction of softening and grain size by linear regression essentially non-applicable. However, a new approach has been applied using ANN in processing simulation and control, solving complex problems without the need to postulate a model with parameter identification [19]. In this model, an advanced statistical information system with a hierarchical structure of neurons grouped in different layers (input layers, hidden layers and output layers) is applied, with characteristics of adaptive learning parameters, which can not only make empirical decisions but also generalize to other conditions. A disadvantage of this method, however, is that it does not provide a reasonable physical explanation for the model, so it is adapted to fit the Arrhenius–Avrami formalism.

The microstructural aspects of static softening were determined from micrographs of samples taken from the test specimens. These specimens were cooled in water after the time between passes ($t_p$), embedded in Bakelite, sanded, polished and electrolytically etched with 65% nitric acid, under a potential of 1.0 mV dc. This treatment revealed the grain boundaries and the presence of precipitates, as well as their shapes, sizes and grain boundary distribution. The process of recrystallization was examined in an optical microscope (Olympus BX51 TRF). Changes in mean grain size ($d_s$) and recrystallized fraction ($X_s$) were statistically analyzed using Image-Pro Plus software, scanning electron microscopy (Phillips XL30 FEG) and transmission electron microscopy (FEI Tecnai). In

addition, EDS microanalysis and X-ray diffraction (X'Pert PRO) were employed to investigate the strain-induced interactions between recrystallization and precipitation in the time between passes ($t_p$).

## 3. Results and Discussion

### 3.1. Continuous Isothermal Stress–Strain Curve

Figure 2 illustrates the stress–strain curves produced by continuous isothermal hot compression tests at different temperatures. Note that these curves initially present a stage of work hardening (WH) in which stress increases continuously, particularly at low temperatures. The flow curves at 1100–1150 °C show closer stress levels. This is due to hot deformation which favors the dynamic softening processes (dynamic recovery—DRV and dynamic recrystallization—DRX) to occur simultaneously and predominate over the work-hardening (WH). Under these conditions, the intense thermal vibration facilitates the diffusion of atoms, the mobility and annihilation of dislocations, contributing to the elimination of dislocations leading to the formation of new grains. The curvature of all stress–strain curves changes as deformation continues, indicating the occurrence of DRX at strain levels below $\varepsilon_c < 0.30$. This can be determined analytically by applying the differentiation method of Poliak & Jonas [20], depicted in Figure 3, whose highest stress level indicates a high WH rate. As deformation increases, the maximum peak strain ($\varepsilon_p$) is reached, increasing slightly in response to temperature. Lastly, there is a stage of dynamic softening, and the stress level gradually decreases with temperature to an intermediate level, showing strong evidence of DRV until the steady state ($\varepsilon_{ss}$) is reached.

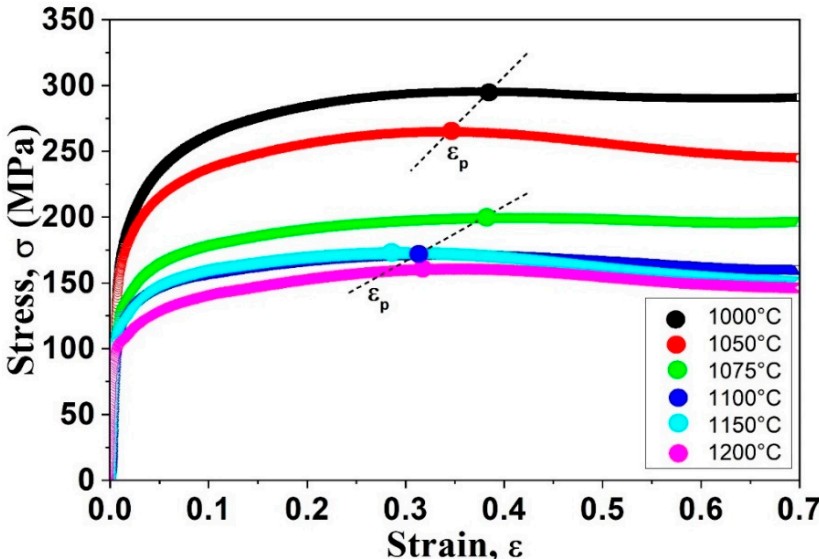

**Figure 2.** Stress–strain curves resulting from continuous isothermal hot compression tests performed at different strain temperatures.

An analysis of the stress–strain curves to determine the critical strain of the onset of DRX ($\varepsilon_c$) under continuous isothermal hot compression indicated that the amount of strain above $\varepsilon = 0.15$ in the double-pass hot compression tests led to static softening by MDRX. Following the onset of DRX, the growth kinetics of nuclei increased after the deformation of the 1st pass was stopped. The remaining material underwent SRV and SRX according to the time between passes ($t_p$), without requiring incubation, since it used the nuclei strained by DRX. After unloading, the material's microstructure underwent a rapid evolution, whose driving force was the elimination of strain-induced dislocations, reducing stored energy. Similar results were reported by Miao, J. et al. [21], who studied the MDRX kinetics of AISI 304/304L, 316/316L steels.

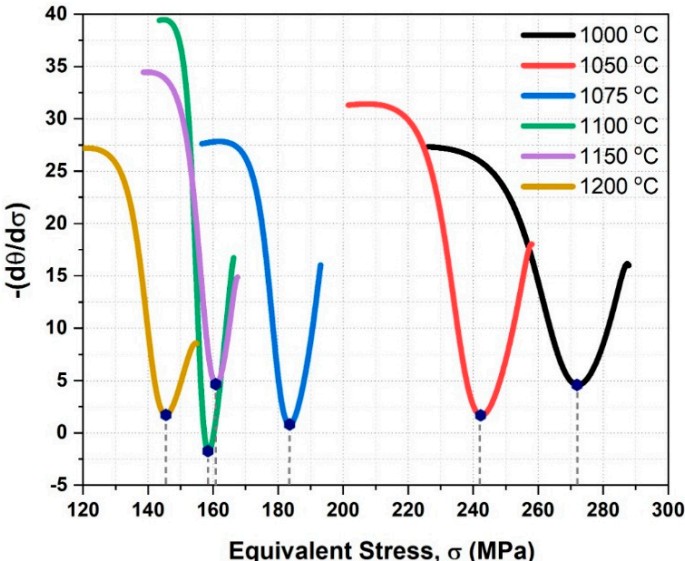

**Figure 3.** Analytical determination of the critical stress of the onset of DRX ($\sigma_c$) in stress–strain curves produced by continuous isothermal hot compression tests from the work hardening rate ($\theta$).

### 3.2. Isothermal Stress–Strain Curves Recorded in Double-Pass Compression Tests

Figures 4 and 5 depict the stress–strain curves obtained from hot compression tests. As can be seen, during the 1st pass ($\varepsilon_1$) the curves show small differences resulting from the heterogeneous distribution of strain and the local temperature gradient, allied to a continuous increase in the stress level, indicating a high work hardening (WH) rate, followed by curving with increasing strain, as a consequence of DRV. Note that, in some conditions, the 1st pass ($\sigma_1$) produces lower stress than the 2nd pass ($\sigma_2$), reflecting the WH caused during the 1st pass, whose driving force does not suffice to completely eliminate dislocations during the time between passes ($t_p$) [22,23]. Furthermore, as the temperature increases, the stress resulting from the 1st ($\sigma_1$) and 2nd passes ($\sigma_2$) decreases rapidly. The difference in stress levels is greater at low temperature, indicating a weak SRV under these conditions, while the stress produced by the 2nd pass ($\sigma_2$) decreases as the time between passes ($t_p$) increases.

With regard to the stress level of the 2nd pass ($\sigma_2$), different unloading stresses were detected, and at $t_p = 1.0$ s the level increased rapidly to that of the 1st pass ($\varepsilon_1$), particularly at $\varepsilon = 0.15$, without significant softening, while the curve of the 2nd pass coincided with the extrapolation of the 1st pass. However, when $t_p > 10$ s, softening was significant and the 2nd pass presented curves with stress levels compatible with the beginning of loading, modifying the shape of the curves. Static softening tended to be significant, causing a decrease in reloading. However, when the time between passes ($t_p$) varied from 10 s to 400 s, the stress level was less reduced, suggesting that the softening process was hindered by an additional mechanism. Lastly, when $t_p > 800$ s, static softening was complete and yield stress was affected by SRV and SRX, whose main mechanism was strain-induced grain boundary migration. In this condition, the $t_p$, temperature, strain, and strain rate significantly affected the behavior of the curves.

An analysis of the curves of the 2nd pass indicates that static softening occurred in all the conditions, followed by a decrease in the degree of WH, especially in conditions of low temperature and shorter $t_p$, indicating that SRX was partial. At longer $t_p$, the curves of the 2nd pass were less pronounced than those of the 1st pass. This indicates that static softening kinetics significantly affected the stress level, depending not only on the strain energy density accumulated during the 1st pass, but also on SRV during $t_p$, which is enhanced with increasing temperature. This caused the stress of the 2nd pass to decrease, albeit above the plateau at which the degree of softening showed a significant increase, probably due to the development of smaller grains resulting from SRX. Moreover, as the

temperature increased in each pass, the stress level declined. It should be noted that the curves of the 2nd pass showed several inversions the longer the $t_p$, which were attributed to the conditions of strain and possibly an additional mechanism that interfered with the kinetics of static softening.

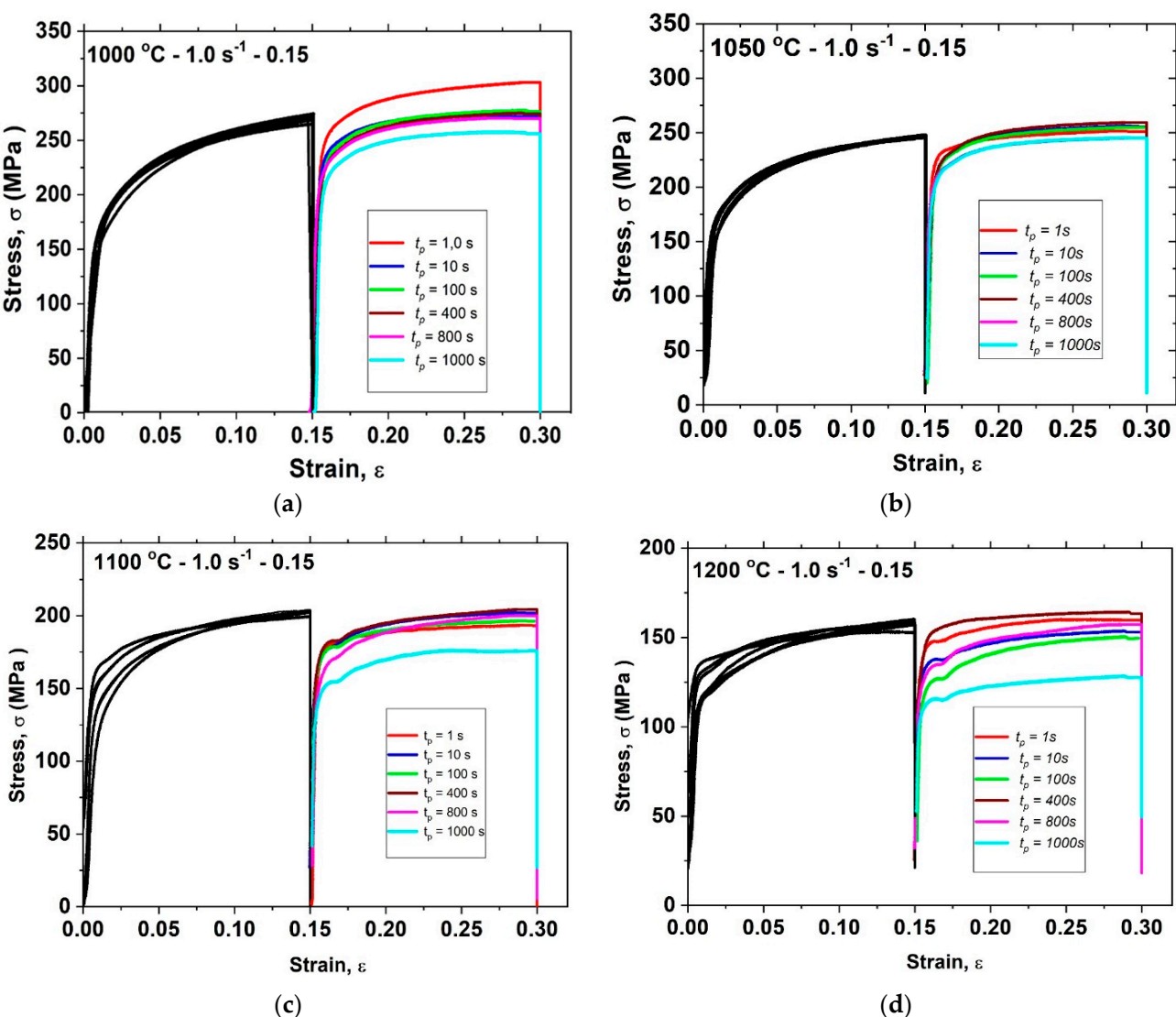

**Figure 4.** Stress–strain curves resulting from double-pass hot compression tests when $\varepsilon = 0.15$ at (**a**) 1000 °C, (**b**) 1050 °C, (**c**) 1100 °C and (**d**) 1200 °C.

Tables 2 and 3 list the softening fractions ($X_s$) calculated from the stress–strain curves obtained in double-pass hot compression tests, $\varepsilon = 0.15$ and $\varepsilon = 0.30$, respectively, using the phenomenological methods of 0.2% offset, 2.0% offset and EMS, according to Equations (1) and (2). Note that the 0.2% offset method considers that 20% of softening is due to SRV; however, in some conditions, this steel presents more than 40% of static softening in less than 1.0 s, which suggests that the softening fraction ($X_s$) is overestimated. This finding is also an indication that the DRV rate is high, which may affect loading after the 1st pass. In these Tables, also note that the EMS method predicts higher levels of reloading stresses ($\sigma_y$), and therefore, a lower softening fraction ($X_s$) than the offset methods. This is evidence of the superior results of SRX kinetics, since they exclude the effects of SRV in the calculation of the softening fraction ($X_s$). This softening between passes is important in the hot forming process, affecting the amount of loading in each pass and hence the microstructure of the material.

**Table 2.** Softening fractions ($X_s$) calculated from the stress–strain curves obtained from double-pass hot compression tests when ε = 0.15.

| | | | Softening Fraction, $X_s$ (%) | | | | | |
|---|---|---|---|---|---|---|---|---|
| **T (°C)** | **ε** | **Method** | Time between Passes, $t_p$ (s) | | | | | |
| | | | **1.0** | **10** | **100** | **400** | **800** | **1000** |
| **1000** | | **0.2%** | 28 | 53 | 68 | 77 | 89 | 99 |
| | | **2%** | 26 | 47 | 63 | 71 | 84 | 96 |
| | | **EMS** | 25 | 44 | 54 | 68 | 78 | 92 |
| **1050** | | **0.2%** | 17 | 40 | 48 | 35 | 72 | 73 |
| | | **2%** | 52 | 36 | 43 | 33 | 66 | 68 |
| | | **EMS** | 51 | 36 | 42 | 32 | 62 | 66 |
| **1100** | **0.15** | **0.2%** | 58 | 57 | 65 | 56 | 87 | 95 |
| | | **2%** | 51 | 52 | 62 | 49 | 79 | 91 |
| | | **EMS** | 48 | 51 | 57 | 47 | 75 | 90 |
| **1200** | | **0.2%** | 48 | 60 | 51 | 48 | 67 | 91 |
| | | **2%** | 45 | 58 | 48 | 37 | 64 | 89 |
| | | **EMS** | 42 | 54 | 46 | 33 | 56 | 88 |

**Table 3.** Softening fractions ($X_s$) calculated from the stress–strain curves obtained from double-pass hot compression tests when ε = 0.30.

| | | | Softening Fraction, $X_s$ (%) | | | | | |
|---|---|---|---|---|---|---|---|---|
| **T (°C)** | **ε** | **Method** | Time between Passes, $t_p$ (s) | | | | | |
| | | | **1.0** | **10** | **100** | **400** | **800** | **1000** |
| **1000** | | **0.2%** | 49 | 67 | 56 | 61 | 89 | 92 |
| | | **2%** | 51 | 56 | 54 | 65 | 82 | 86 |
| | | **EMS** | 46 | 54 | 52 | 60 | 70 | 75 |
| **1050** | | **0.2%** | 41 | 44 | 38 | 74 | 62 | 90 |
| | | **2%** | 42 | 43 | 37 | 69 | 62 | 87 |
| | | **EMS** | 39 | 39 | 33 | 65 | 59 | 87 |
| **1100** | **0.30** | **0.2%** | 39 | 46 | 49 | 55 | 72 | 76 |
| | | **2%** | 36 | 44 | 47 | 51 | 69 | 77 |
| | | **EMS** | 35 | 42 | 43 | 51 | 65 | 74 |
| **1200** | | **0.2%** | 45 | 66 | 53 | 65 | 80 | 100 |
| | | **2%** | 42 | 62 | 49 | 63 | 75 | 92 |
| | | **EMS** | 41 | 60 | 48 | 59 | 74 | 91 |

A comparison of the softening fractions ($X_s$) calculated by different methods revealed the absence of linearity, probably due to the higher fraction of SRV included in the overestimated softening fraction ($X_s > X_{SRX}$), potentiated by increasing temperature. Moreover, this comparison provides a good prediction of the 50% softening ($t_{0.5}$) values based on the $X_s$ vs. $t_p$ curves. This abnormal static softening was attributed to the significant contribution of SRV, given the material's moderate stacking fault energy ($\gamma_{sfe}$), segregation of Nb-N solute with dislocations during the time between passes ($t_p$) and the presence of fine strain-induced precipitates that hinder grain boundary mobility. For comparison purposes, similar studies of AISI 304, 304L, 316, 316L and 316LVM steels [24,25] have not reported delays in the kinetic behavior of static softening under these conditions.

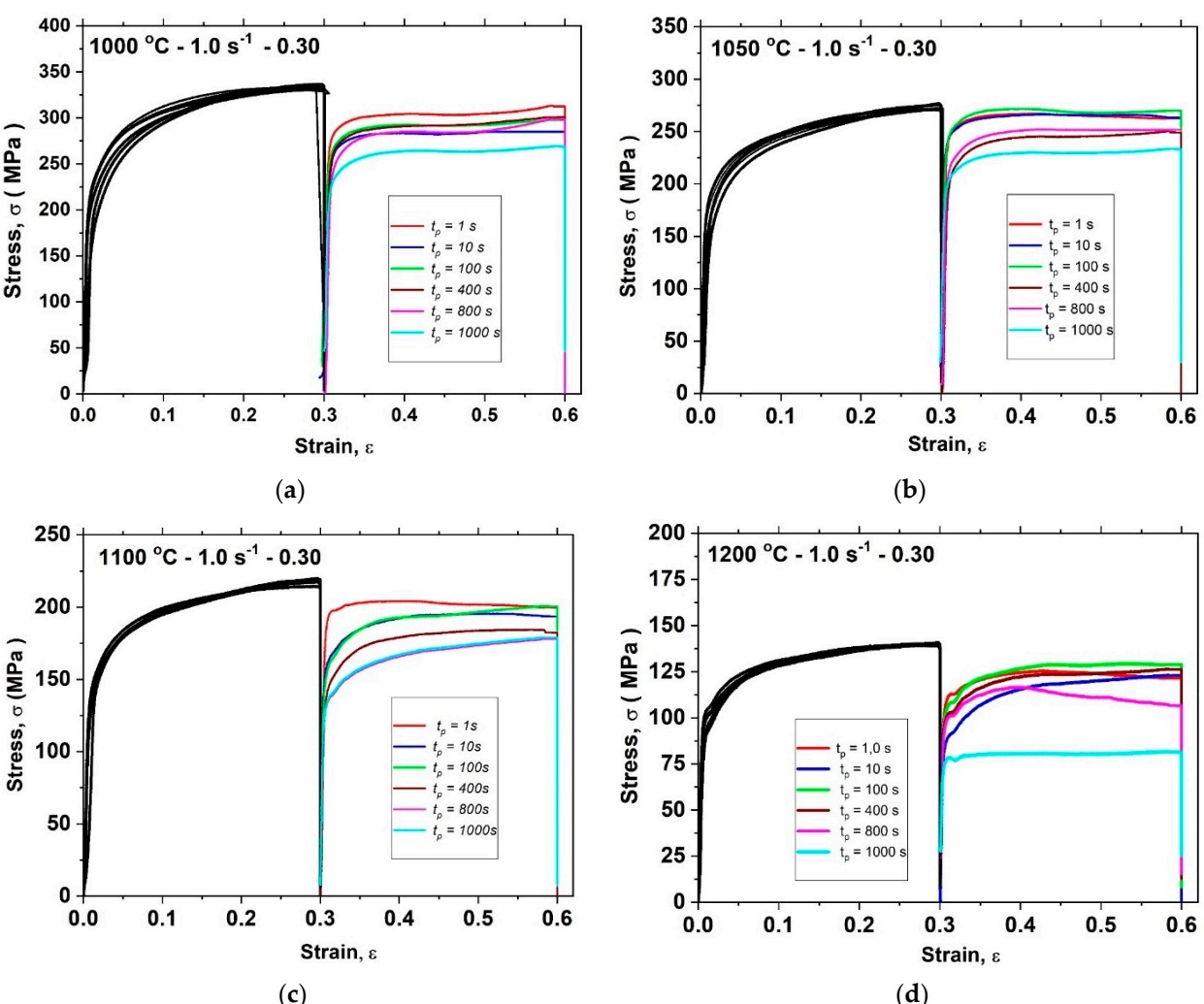

**Figure 5.** Stress–strain curves resulting from double-pass hot compression tests when $\varepsilon$ = 0.30 at (**a**) 1000 °C, (**b**) 1050 °C, (**c**) 1100 °C and (**d**) 1200 °C.

*3.3. Dependence of $X_s$ on $t_p$*

Figure 6 illustrates the evolution of the softening fraction ($X_s$) as a function of time between passes ($t_p$), using the 0.2% offset, 2.0% offset and EMS methods under two strain conditions: (a) $\varepsilon$ = 0.15 and (b) $\varepsilon$ = 0.30. As a whole, the curves show two distinct behaviors, with transition occurring at close to 1050 °C. The area above the curves shows the formation of plateaus extending over a long period of time, delaying the static softening kinetics by up to approximately $t_p$ = 800 s. From there on, the softening fraction ($X_s$) resumes its accelerated growth, suggesting that softening initially takes place through intense SRV, up to $X_s$ = 40% at $t_p$ = 1.0 s, without a lag time prior to effective precipitation and without delaying the kinetics of SRX. As can be seen, no plateau is formed in the condition of 1000 °C and $\varepsilon$ = 0.15, although the curve becomes smoother and SRX continues, with the stored energy lower than the anchoring effect of precipitates, even during long $t_p$. These results indicate that the softening phenomenon becomes more marked with increasing $t_p$, as a result of the decrease in local dislocation density, followed by a significant reduction in the WH rate.

In Figure 6, note the increase in the rate of static softening ($X_s$) by SRV in response to increasing temperature, which accounts for almost 40% at the beginning of $t_p$. As the $t_p$ increases, partial softening continues, since the softening fraction ($X_s$) remains approximately 40–60% for a long time, maintaining the formation of the plateau instead of acquiring the sigmoidal shape characteristic of the Avrami formalism for microalloyed

steels, as illustrated in Figure 6a,b [26]. This condition is believed to be due to the formation of some additional of mechanism that increases resistance, such as the formation of strain-induced precipitates that inhibit grain boundary mobility and delay the static softening kinetics, starting at $t_p = 1.0$ s. Under the lower strain ($\varepsilon = 0.15$), this interaction occurs at $t_p > 10$ s, and is a thermally activated phenomenon.

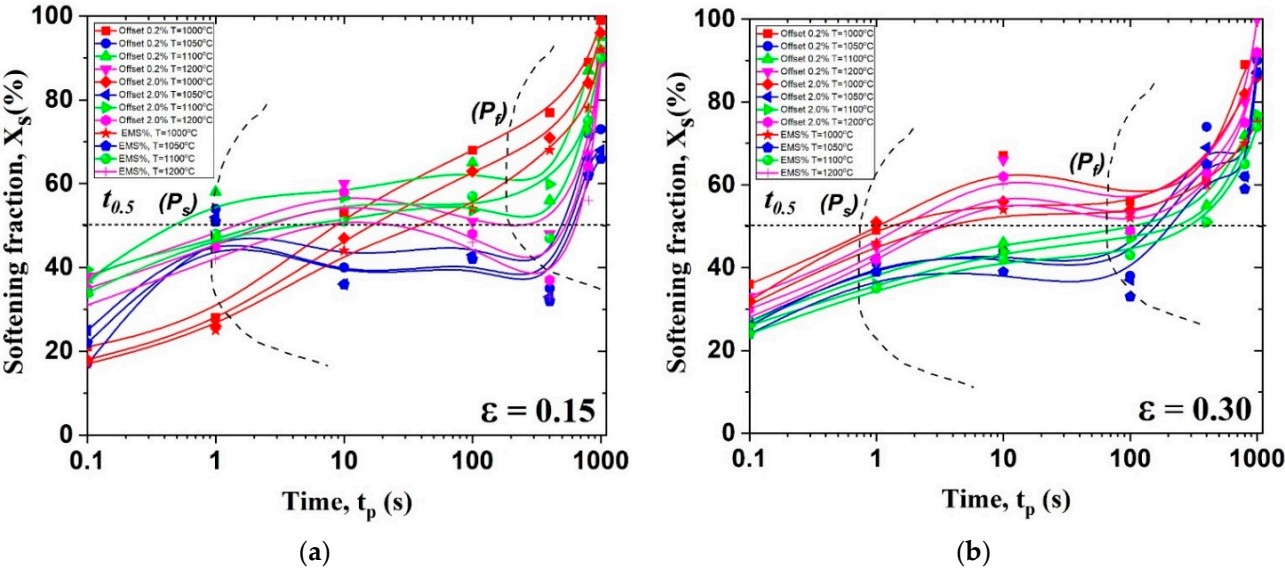

**Figure 6.** Dependence of the softening fraction ($X_s$) on time between passes ($t_p$), calculated according to the 0.2% offset method: (**a**) $\varepsilon = 0.15$, and (**b**) $\varepsilon = 0.30$.

Starting at $t_p = 1.0$ s, the softening mechanism shows a transition, with the gradient of the $X_s$ vs. $t_p$ curve changing, followed by the partial suppression of softening. This suppression may be a consequence of the effects of strain-induced precipitates that inhibit the nucleation and growth of recrystallized grains, as reported by Medina and Mancilla [27]. Those authors, who studied Nb-V microalloyed steels, reported that a plateau appeared in the $X_s$ vs. $t_p$ curves within this temperature range. This suggests that the temperature range in which the plateau is formed should increase the higher the temperature and strain, and that the dissolution of these precipitates in the time between passes ($t_p$) can be controlled, such as the concentration of Nb and N in solution, which increases with the dissolution of precipitates at higher temperatures. Thus, during the formation of the plateau, a competition occurs between the driving force for SRX and the anchoring force of precipitates, inhibiting SRX until the grains reach a stage of coalescence and evolution. Therefore, when the volumetric fraction of precipitates is high, the plateau extends for a long time between passes ($t_p$), since the high density of precipitation causes its kinetics to control volume diffusion, delaying the onset of coalescence, which in turn delays the resumption of softening.

With increasing time between passes, i.e., $t_p > 400$ s, note that the softening fraction ($X_s$) resumes its growth, indicating that the coalescence of precipitates favors grain boundary mobility. However, when $t_p > 100$ s, the softening fraction ($X_s$) at T = 1050 °C remains constant, suggesting that SRX did not proceed. Lastly, at T > 1100 °C and $t_p > 800$ s, the plateau is not maintained, indicating that the relationship between the size and volumetric fraction of precipitates is insufficient to prevent SRX. It can be concluded that, in these conditions, SRV has more marked effects on the level of stress, and hence, on the softening fraction ($X_s$) than SRX. The reason for this is that, although the phenomenon of SRX explains variations in the softening fraction, SRV begins before SRX and, in this material with a high percentage ($X_{srv} > 30\%$), it is attributed to the moderate value of $\gamma_{sfe}$. In addition, the occurrence of SRX requires a period of time between passes, and as a result, a large number of dislocations are consumed in the early stages of static softening, since the driving force

for SRX is weak, with a low volumetric fraction. Therefore, SRV and SRX affect the behavior of stress–strain curves.

### 3.4. Determination of 50% Softening Time (t$_{0.5}$) and the Avrami Exponent (n)

Based on the X$_s$ vs. $t_p$ softening curves (Figure 6), the time spent to reach 50% of softening ($t_{0.5}$) is estimated using the Avrami method [28]. On the other hand, the logarithmic linearization of the X$_s$ vs. $t_p$ curve was applied to estimate the Avrami exponent ($n$), with the value of $n$ determined by the gradient of the $\ln[\ln[1/(1 - X_s)]]$ vs. log ($t/t_p$), curve (Figure 7). Note that the lines are not parallel and that, at the same temperature, the $n$ values are independent of the strain, which varied from 0.10 < $n$ < 0.28, and was dependent on the temperature, amount of strain, initial grain size and composition. These values are lower than those reported in the literature for 300 M steel ($n$ = 0.34), AISI 304 (0.19 < $n$ < 0.38), and AISI 316LN ($n$ = 0.46) stainless steels [29,30].

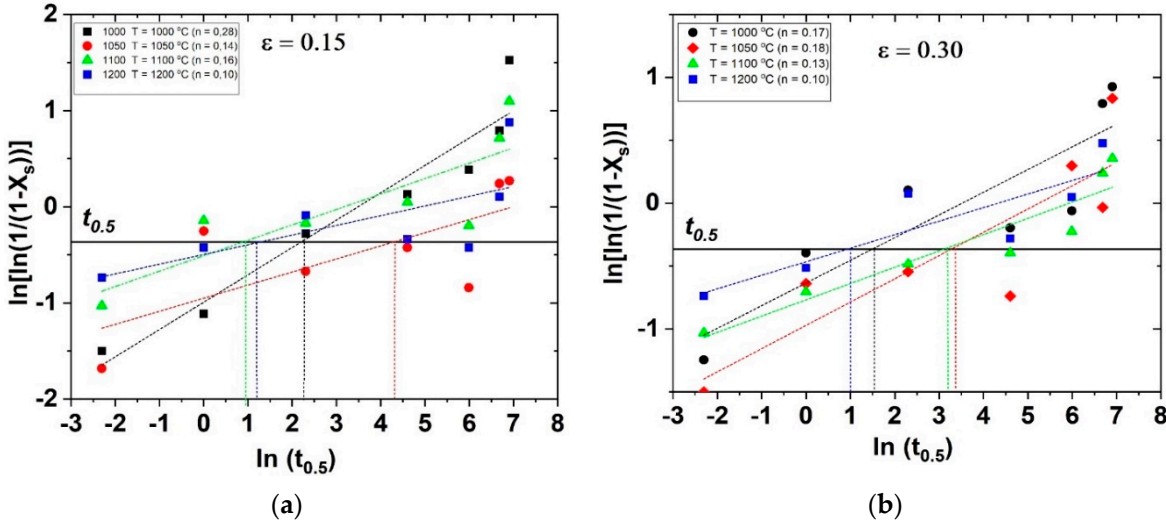

**Figure 7.** Calculation of the Avrami exponent ($n$) in the double-pass hot compression tests: (**a**) $\varepsilon$ = 0.15 and (**b**) $\varepsilon$ = 0.30. (1000 °C < T < 1200 °C and 1.0 < $t_p$ < 1000 s).

The different values of $n$ depend on the material's deformation conditions and microstructural responses, with a high SRV rate and few sites of SRX, delaying the nucleation and growth of recrystallized grains, as reported by E.J. Giordani [31], who studied the competing forces of static recovery and static recrystallization, SRV-SRX, of this steel. An analysis of Figure 6 suggests that in softening by MDR$X$ ($\varepsilon$ = 0.30), whose driving force is dislocation density due to DRX, the nucleation rate is lower than in SRX ($\varepsilon$ = 0.15), suggesting a low value of $n$. Moreover, in stainless steels, the value of $n$ tends to decrease as grain size increases.

### 3.5. Proposal of Constitutive Equations for t$_{0.5}$ and n

Based on the experimental values of time to reach 50% static softening ($t_{0.5}$), indicated on the X$_s$ vs. $t_p$ curves, the average activation energy for static softening ($Q_s$ = 513 kJ/mol) was estimated by linear regression, considering the slope ($Q_s/R$ = 0.617) of the ln ($t_{0.5}$) vs. $1/T$ (K) curve, as shown in Figure 8. This value is above those reported in the literature for AISI 304 ($Q_s$ = 350 kJ/mol) and AISI 316 ($Q_s$ = 416 kJ/mol) steels [32]. These high values are due to the presence of the elements Nb and N under the solute effect [32,33]. It should be noted that in these conditions, the static softening kinetics is faster the higher the deformation temperature. In addition, the time to reach 50% of softening ($t_{0.5}$) decreases the greater the amount of deformation, and this behavior is inverse only in the condition of T = 1050 °C, which is another indication of the effect of delay on the static softening kinetics.

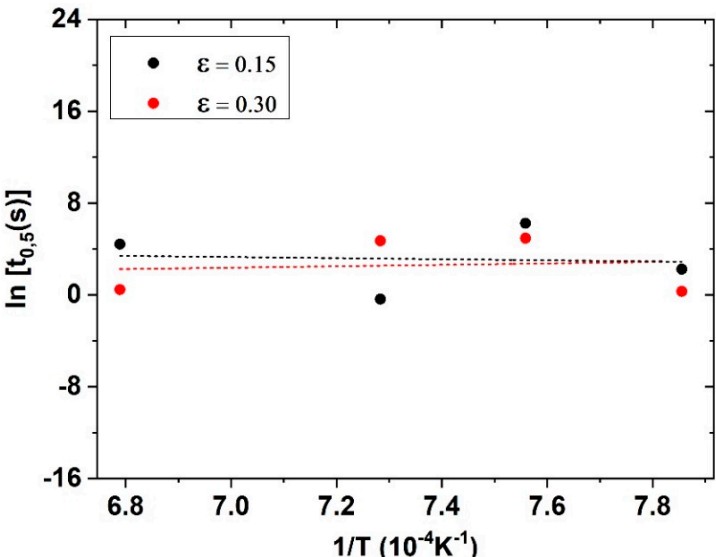

**Figure 8.** Relationship between 50% softening time ($t_{0.5}$) and the inverse of absolute temperature, determination of activation energy for static recrystallization ($Q_s$).

By applying the natural logarithm, one has:

$$\ln t_{0.5} = \ln A + r \ln d_o + q \ln \varepsilon + p \ln Z + \frac{Q_s}{RT} \tag{5}$$

The dependence on strain ($\varepsilon$), grain size ($d_o$) and Z parameter ($Z = \varepsilon.\exp(587\,\text{kJ/mol}/RT)$) was determined similarly, according to strain conditions, Figures 8 and 9. The values of $r$, $q$ and $p$ were found to be 2.0, 0.255, and $-0.817$, respectively. These values vary strongly as a function of temperature, which makes it difficult to develop a general model for predicting time to achieve 50% softening ($t_{0.5}$). The results indicate that the Avrami exponent ($n$) gradually decreased along with the deformation temperature and that the Z parameter was not adequate to evaluate the softening kinetics of MDRX, since the time between passes ($t_p$) was not included directly in the analysis, unlike the strain rate, temperature and activation energy.

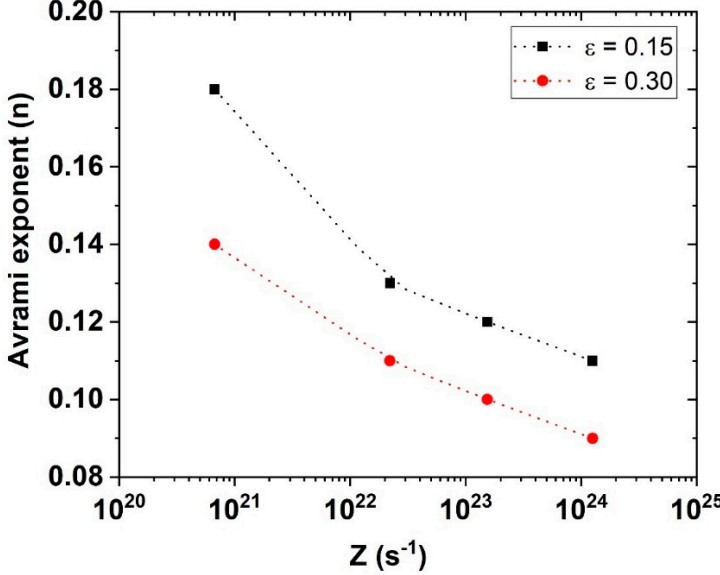

**Figure 9.** Dependence of the Avrami exponent ($n$) on the Zener-Hollomon ($Z$) parameter in the conditions of double-pass hot compression.

The static softening kinetics of ASTM F-1586 steel can be described using multiple variable regression analysis, as expressed in Equations (7) and (8). Multipass thermomechanical processing is common in hot deformation. These operations can only be controlled when the microstructures and softening mechanisms that occur during passes are known, and these constitutive equations are used to predict such behaviors in formalizing the softening fraction ($X_s$). However, the absence of a standard method to evaluate the softening rate leads to significant variations in the determination of the experimental constants of $t_{0.5}$ expressed in Equation (8).

$$n = 2.47 \times Z^{-0.060} \tag{6}$$

$$X_s = 1 - exp\left[-0.693\left(\frac{t}{t_{0.5}}\right)^{0.14}\right] \tag{7}$$

$$t_{0.5} = Ad_o^{2.0}\varepsilon^{0.255}Z^{-0.817}exp\left(\frac{513\ \text{kJ/mol}}{RT}\right) \tag{8}$$

*3.6. Proposal of an Artificial Neural Network (ANN)*

ANNs have been frequently used to model the mechanical properties and hot deformation behavior of steels in different metallurgical conditions, with good results described in the literature [34,35]. The main advantage of using an ANN is its versatility in developing problems that are complex by traditional computational methods. However, the successful application of an ANN model is strongly dependent on the availability and quality of the dataset and variables of the problem, with adjustable training and testing, since such modeling does not provide a physical view of the problem [36]. In this work, the ANN creation project is structured on the collection of data from the 0.2% offset, 2.0% offset and EMS methods and on constitutive equations (Equations (7) and (8)), which are formalized using the Arrhenius–Avrami methodology. Our proposal is to develop an ANN that allows one to estimate the static softening fraction ($X_s$) and average size of recrystallized grains ($d_s$) under different conditions of hot deformation, according to the schematic diagram depicted in Figure 10.

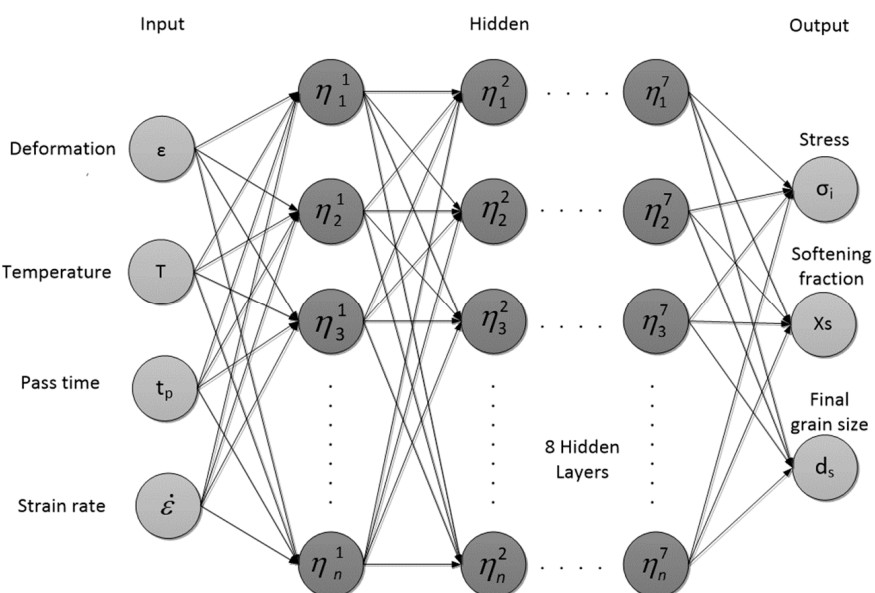

**Figure 10.** Schematic diagram of the architecture of the proposed ANN used for the computational simulation of double-pass hot compression tests.

Figure 11 illustrates adaptive training involved in the development of the ANN to calculate the softening fraction ($X_s$) as a function of the number of samples, according to the

offset 0.2%, offset 2.0% and EMS methods. The synaptic weights of these methods are modifiable according to the learning algorithm, having a learning rate of 0.8 in 10,000 epochs, depending on the input signal ($T$, $\varepsilon$, $\sigma_i$, $t_p$), and the output values ($X_s$, $d_s$) are associated with the supervised backpropagation learning response used in training the structure of the direct multilayer network with static neurons. After an input pattern with three parameters was applied to stimulate the elements of the 1st layer, the pattern was repeated through four hidden layers containing six, fourteen, twenty and ten neurons, respectively, and an output layer. The latter included the softening fraction ($X_s$), and the activation function of the hyperbolic tangent type was adopted in the first three hidden layers, a sigmoid function in the fourth layer, and a linear function in the output layer.

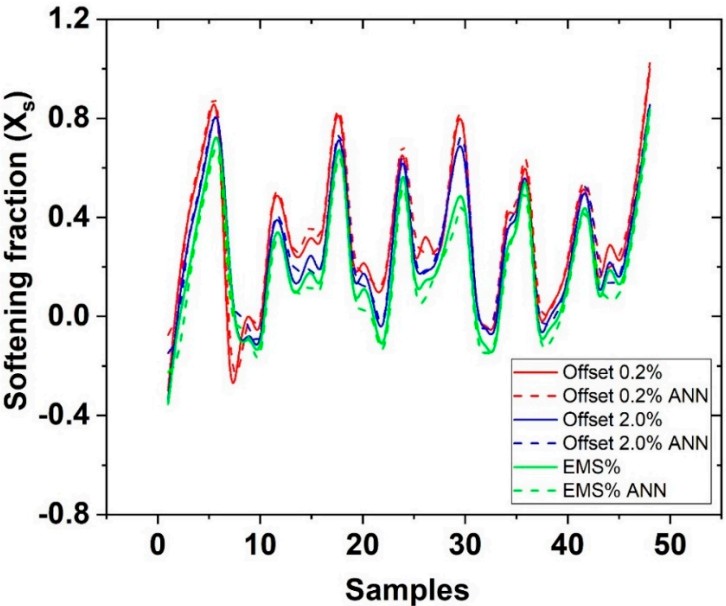

**Figure 11.** Training the ANN to calculate the softening fraction ($X_s$) as a function of the number of samples, using the 0.2% offset, 2.0% offset and EMS methods.

This output pattern was then compared to the desired one and an error signal was calculated. This error signal was then backpropagated to the elements of the hidden layers, with each element receiving only a portion of the total error signal, proportional to the relative contribution of each element in forming the original output. This process was repeated in each layer until each element of the ANN received an error signal describing its relative contribution to the total error. Based on the error signal, the weights of the connections were updated so that the ANN would converge to a state that allowed all the patterns of the training set to be encoded. The greater the number of input data, the better the ANN's prediction performance.

### 3.7. Validation of the Proposed ANN Using Constitutive and Analytical Methods

Figures 12 and 13 show the validation of the ANN model applied in the calculation of the softening fraction ($X_s$) compared to the constitutive and analytical models described in the literature [37]. The ANN's numerical results are consistent with the experimental data and existing literature about hot deformation, as indicated by the correlation coefficient, $R^2 = 0.94$ depicted in Figure 14. Such consistency suggests that this ANN can provide a good description of the static softening process of this steel under various conditions of hot deformation. It should be kept in mind that a high value of $R^2$ does not always indicate better performance of the ANN, since the values tend towards the estimated maximum and minimum. Therefore, the mean absolute error ($\Delta$) is a good statistical parameter of the predictability of these models.

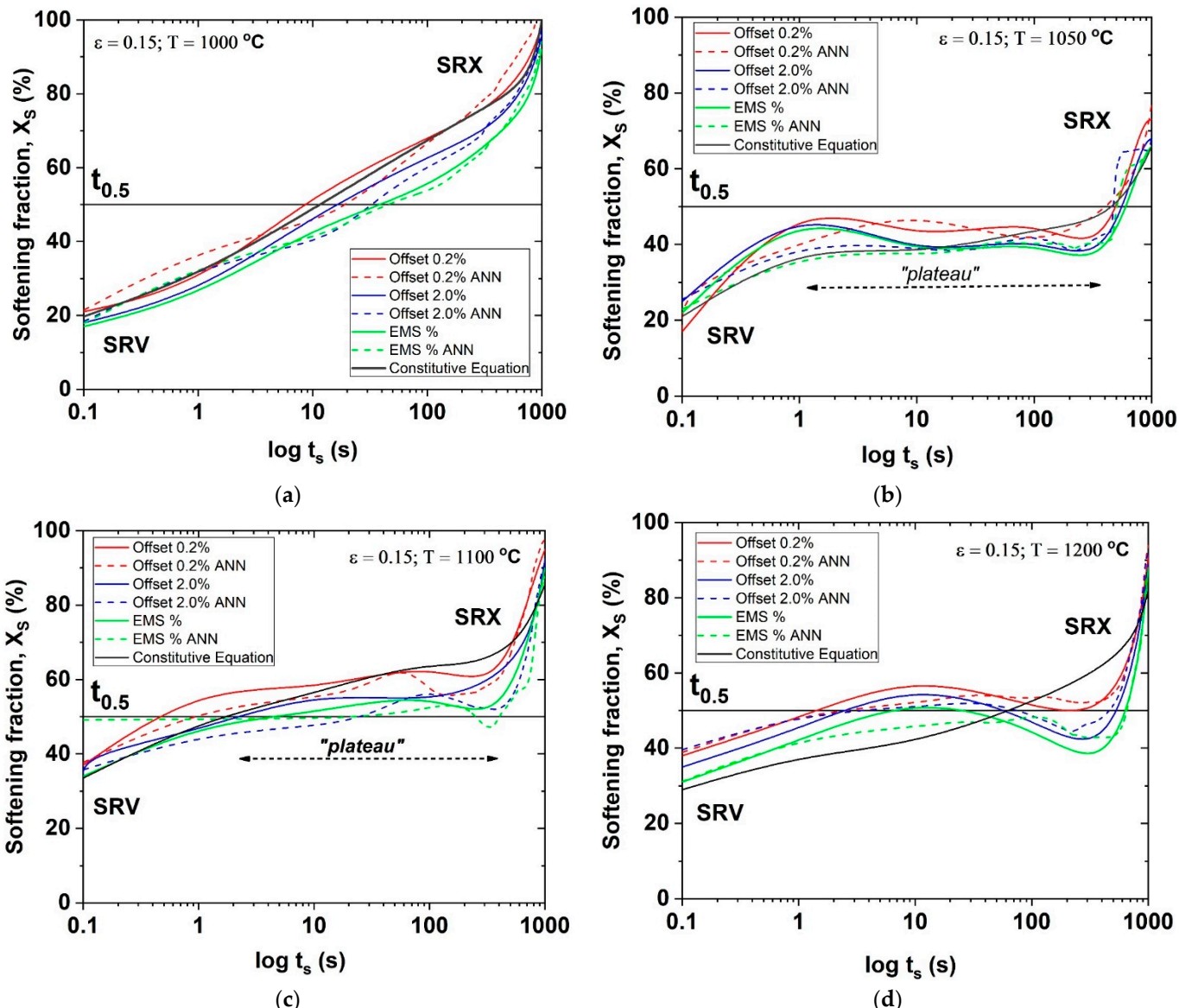

**Figure 12.** Validation of the ANN model employed to calculate the static softening fraction ($X_s$) under the conditions of $\varepsilon = 0.15$ at (**a**) 1000 °C, (**b**) 1050 °C, (**c**) 1100 °C and (**d**) 1200 °C.

The results indicate that the mean absolute errors (Δ) of the constitutive models 0.2% offset (Δ = 2.13), 2.0% offset (Δ = 1.92) and EMS (Δ = 2.13) are greater than those obtained from the ANN model (Δ = 0.85%, 0.61%, and 0.56%), respectively. A comparison of the results indicates that the ANN model has a higher $R^2$ value and lower Δ value; hence, the constitutive relation developed using the ANN is more accurate for the study of thermomechanical processing under these conditions than the Arrhenius–Avrami constitutive model. Therefore, the ANN model has an excellent ability to predict the static softening of ASTM F-1586 steel under hot compression. Li et al. [38] also used the ANN model to predict the hot behavior of a low-alloy steel and obtained satisfactory results, as did Lucon et al. [39], who used the ANN methodology to compare the recrystallization kinetics of three specific materials: aluminum, pure iron and IF steel, reported in the literature, obtaining a good approximation between the data and an acceptable error of 0.0001. More recently, Feng et al. [15] reported promising results obtained with the ANN methodology for AISI 304 steel using thirty-six samples with a satisfactory mean squared error (Δ).

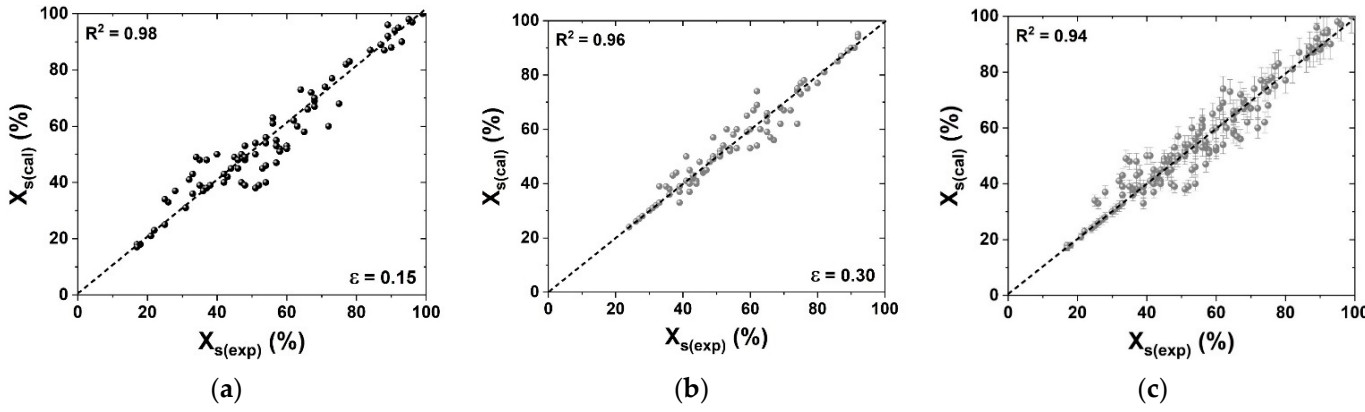

**Figure 13.** Validation of the ANN model employed to calculate the static softening fraction ($X_s$) under the conditions of $\varepsilon = 0.30$ at (**a**) 1000 °C, (**b**) 1050 °C, (**c**) 1100 °C and (**d**) 1200 °C.

**Figure 14.** Correlation between the experimental softening fraction $X_{s(exp)}$ and that predicted by the ANN $X_{s(cal)}$: (**a**) $\varepsilon = 0.15$, (**b**) $\varepsilon = 0.30$ and (**c**) all the conditions.

It is worth noting that the Arrhenius–Avrami method applied here does not depend explicitly on the effect of strain-induced precipitates on the softening kinetics. Therefore, there is no agreement with the formation of plateaus that temporarily halt the softening fraction ($X_s$) in the curves, showing deviations in this region without the typical sigmoidal shape, not to mention the stress instabilities under high hot compression temperature. Nevertheless, it was possible to indirectly observe the delay in the static softening kinetics, with a low Avrami exponent ($n$), which is strong evidence of the delay in recrystallization kinetics.

Note that the proposed ANN model can be used to map this steel's work hardening and static softening under different strain conditions, suggesting that this ANN can easily describe the nonlinear relationship between the stress level ($\sigma_i$), softening fraction ($X_s$) and grain size ($d_s$) of ASTM F-1586 steel. However, analyses of the proposed models show some underestimated predictions of the value of $X_s$ shown in Figures 12 and 13, which does not consider the effective role of SRV and precipitation in the material's softening kinetics. Another underestimated prediction is the effect of relaxation on hot compression machine unloading during long times between passes ($t_p$), which tends to exhibit less resistance during reloading in the 2nd pass, even during short times between passes. Studies in the literature [40] on the microstructural evolution of stainless steels subjected to double-pass compression indicate that overestimating the softening fraction ($X_s$) can be attributed to subgrain growth, with a balance of the softening mechanisms according to the strain parameters.

### 3.8. Microstructural Features

The recrystallized fraction was calculated by metallographic examination using optical (OM) and scanning electron (SEM) microscopy, which differentiated the SRV and SRX grains according to their microstructural features, such as the appearance of grain boundaries, grain size, grain shapes, and necklace formations at grain boundaries. Thus, during the time between passes ($t_p$) at 1000 °C, the microstructure showed evidence of partial SRX ($X = 16\%$), with hardened, elongated grains and necklace formation with low Avrami exponent ($n$), directly affecting the static softening kinetics, as depicted in Figure 15a,b. Note that the microstructure changed during the time between passes ($t_p$) and the low value of $n$ was attributed to intense SRV, as well as formation and growth of subgrains at the elongated grain boundaries and the presence of strain-induced precipitates that inhibited the evolution of SRX, hindering the kinetics and forming plateaus on the $X_s$ vs. $t_p$ curves, as shown in Figure 6.

Thus, SRX occurs only when there is sufficient time for grain nucleation and growth after hot deformation, which is an important phenomenon that alters grain size and distribution. Given that SRX is a thermally activated process, atom diffusion will be greater at high temperatures, causing grain boundary migration, although the grain size at 1000 °C is smaller than at 1200 °C. This suggests that the stored energy is insufficient to trigger full SRX at low temperatures, indicating the existence of a critical temperature range within which to isolate SRX. These microstructural findings are consistent with the behavior of plastic yield stress and with the calculations of softening fractions under different deformation conditions.

During hot deformation, the microstructure is unstable, with concurrent hardening and softening, as well as during unloading between passes. This softening fraction ($X_s$) is responsible for load reduction and grain refinement, depending on the SRV-SRX ratio, whose extent is governed by deformation and material conditions, such as moderate $\gamma_{sfe}$, the presence of Nb-N solute and fine precipitates, which affect grain boundary mobility. The precipitates coalesce and the SRX fronts are released only after a long $t_p$ at a high temperature, T > 1100 °C, developing refined, straighter and equiaxed grain boundaries distributed uniformly in the matrix with full SRX ($X > 90\%$). The importance of the contribution of static softening cannot be overstated, given the need to avoid microstructures of mixed grains resulting from partial SRX, which develop as a function of temperature, amount of strain and $t_p$.

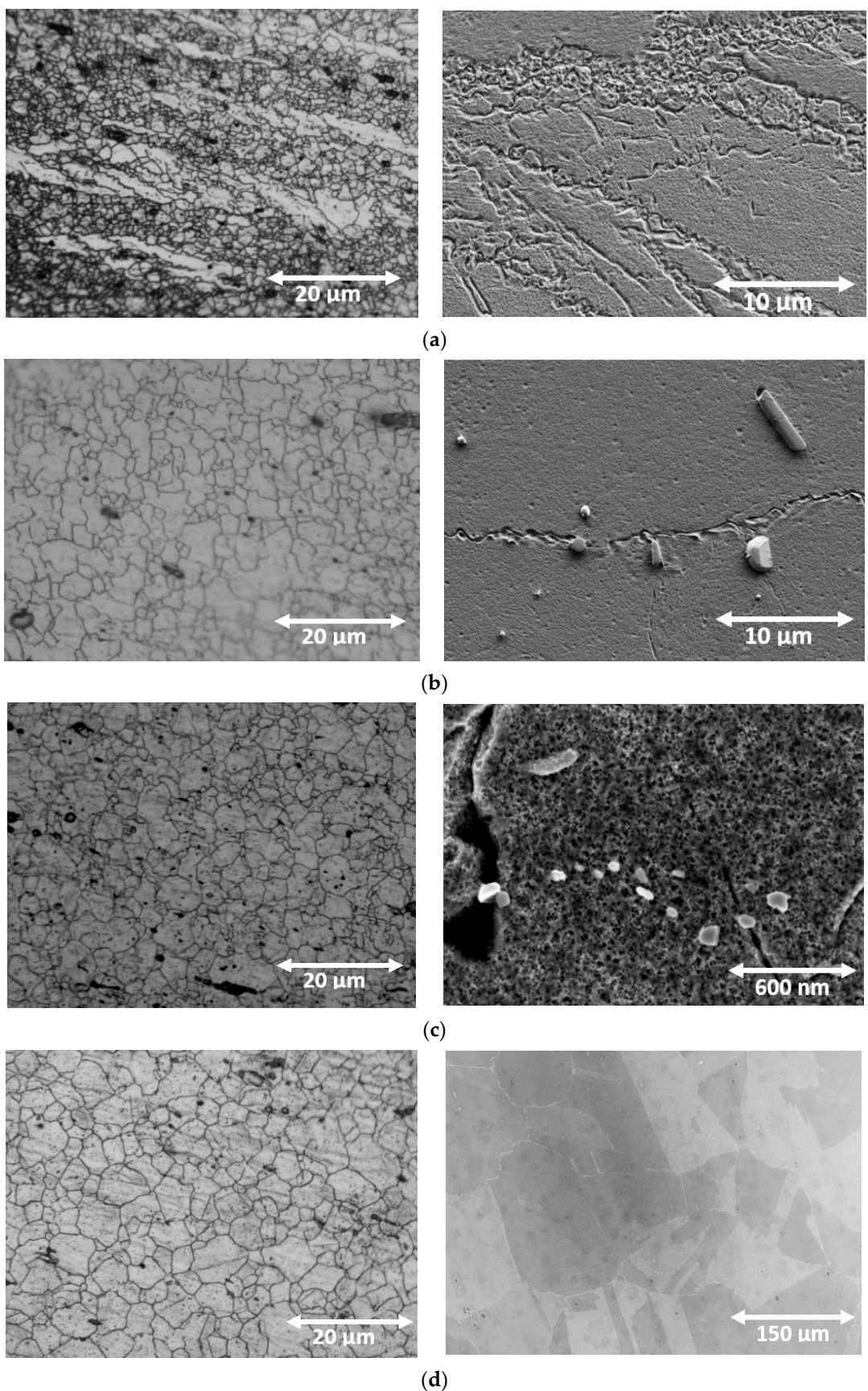

**Figure 15.** Metallographic analysis by OM and SEM of static softening in the conditions of (**a**) 1000 (X = 16%), (**b**)1050 (X = 69%), (**c**) 1100 (X = 75%) and (**d**) 1200 °C for $t_p$ = 800 s (X = 95%).

E.J. Giordani et al. [31], who analyzed the competing forces of SRV and SRX in this steel using scanning electron microscopy (SEM) and electron backscatter diffraction (EBSD), found that 60% of the grains have low angle boundaries, which is a strong indication of SRV, indicating the absence of consonance between softening fraction ($X_s$) and recrystallized fraction (XSRX). In Figure 6, note that 40% of softening occurs in just $t_p$ = 1.0 s at T = 1100 °C, and the microstructure is strongly work-hardened with few SRX grains. Another reason for this discrepancy is the moderate stacking fault energy, $\gamma_{sfe}$ = 69 mJ/m$^2$. Aquino et al. [41] stated that the stacking fault energy of this steel is well above that of other austenitic steels, such as AISI 304/304L ($\gamma_{sfe}$ = 18 mJ/m$^2$) [42] and that this energy hinders the evolution of SRX/MDRX by favoring the action of thermally activated mechanisms and hence the SRV mechanism. The same finding was reported by Gerônimo et al. [43], who studied AISI 316LVM steel, $\gamma_{sfe}$ = 78 mJ/m$^2$ [44]. Lastly, at T > 1100 °C and $t_p$ > 400 s, SRX was in an advanced stage, with fully recrystallized grains resulting from the higher thermal activity and coalescence of precipitates, which inhibited recrystallization, Figure 15d.

Another factor to keep in mind is the influence of solute supersaturation, since the steel was kept at the soaking temperature of 1250 °C for 300 s, and Nb and N supersaturation in the austenite increases with decreasing temperature. Strain-induced precipitation is known to occur due to such supersaturation, which increases the nucleation rate and accelerates precipitation kinetics, and the higher this precipitation the greater its influence on SRX [45]. The supersaturation of Nb in the austenite of this steel at T ≤ 1050 °C is higher than at T ≤ 1100 °C. Thus, precipitation at higher temperatures may be too slow, $t_p$ > 400 s, to inhibit SRX, which is not the case at lower temperatures.

An evaluation of the effect of strain conditions on the softening curves indicated that the softening kinetics was affected by temperature, which steadily altered the shape of the curves. The SRX rate increased with temperature, showing higher grain boundary mobility the greater the number of SRX nuclei and a decrease due to solute drag or precipitate pinning [46]. As for the amount of strain, it was found that the higher the strain the higher the onset of plastic yield stress, thereby increasing the intensity of softening by SRV. In addition, the intensity of precipitation hardening was unaffected by strain, since the plateau levels were similar at the various levels of deformation, which can be explained by the numerous nucleation sites of precipitates even under lower levels of deformation.

Figure 16 shows the microstructures of fine strain-induced *Z*-phase precipitates (CrNbN) (*d* < 200 nm) affecting the grain boundary mobility of the matrix in the condition of T = 1050 °C and $t_p$ = 800 s. The curve in the grain boundary graph indicates that precipitates exert an anchoring effect on the boundaries, inhibiting their mobility. Precipitation occurs more easily at high free energy sites, such as deformation bands and grain boundaries, since interfaces must be created in order to form a new phase within the matrix [47]. Thus, hot deformation induces the formation of fine precipitates, reducing the time required for its onset. This anchoring of precipitates at the grain boundaries hinders SRX grain nucleation, thus temporarily inhibiting its onset and development. However, coarse precipitates can impair mechanical properties, as has been reported in the literature [48].

As for the effect of the amount of strain on the interaction between recrystallization and precipitation, in Figures 12 and 13, note that the plateau begins at shorter incubation times under higher strains. In the condition of *T* = 1000 °C and *ε* = 0.30, the plateau occurs before softening reaches $X_s$ = 50% with $t_p$ = 1.0 s. However, at *ε* = 0.15, softening reaches $X_s$ = 40% above $t_p$ = 10 s, before attaining the plateau. Thus, SRX is hindered by the anchoring effect of strain-induced precipitates, whose size and volumetric fraction affect the anchoring force, inhibiting SRX during hot forming and at the reheating temperature to control grain growth. At higher temperatures, softening proceeds only after a $t_p$, when the precipitates coalesce, and the effect of SRX softening is more pronounced above 50% of softening. Another reason for this divergence in plateau formation is due to the overestimation of time on a logarithmic scale. Investigations [49] have shown that the non-recrystallization

temperature ($T_{nr}$) of this steel is close to 1100 °C, i.e., strain-induced precipitation inhibits SRX, as was observed above 1050 °C.

| OM | SEM | TEM |
|----|-----|-----|

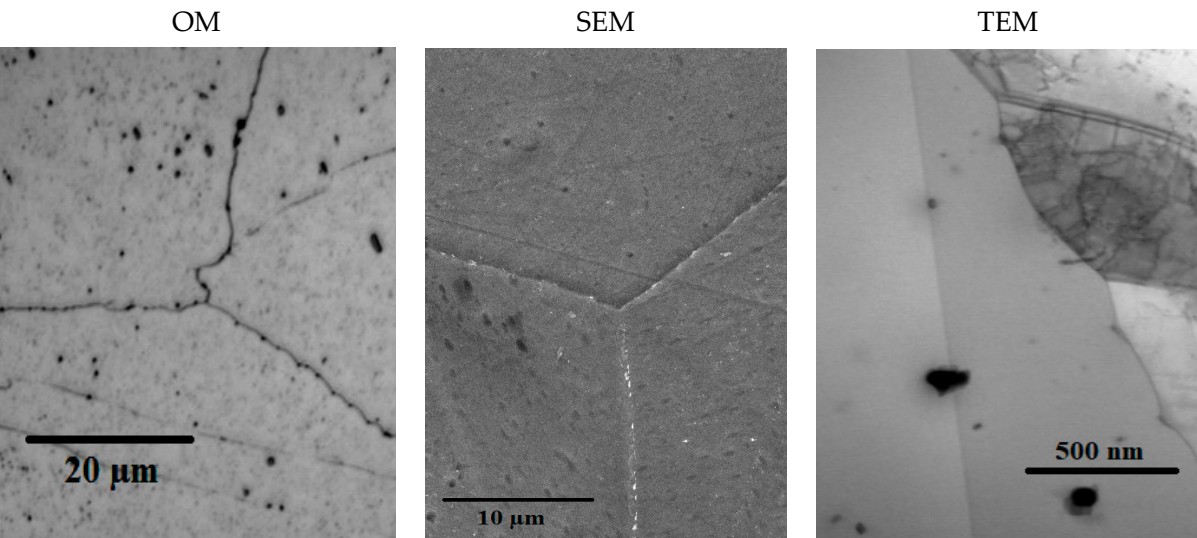

**Figure 16.** Influence of fine strain-induced Z-phase precipitates (CrNbN) ($d < 200$ nm) on grain boundary mobility, T = 1050 °C and $t_p$ = 100 s.

Figure 17 correlates the experimental recrystallized grain size ($d_{s(exp)}$) and the one calculated via ANN ($d_{s(ANN)}$), based on the constitutive model proposed by Tsurekawa, S. [50] and shown in Equation (9). This constitutive relationship (Equation (9)) with a correlation coefficient of $R^2 = 0.96$ indicates that this model is reliable for estimating the average grain size, in order to make forecasts for conditions within this hot working temperature range. The model allows one to associate the variables of stress ($\sigma$), softening fraction ($X_s$) and recrystallized grain size ($d_s$), according to the deformation conditions, to control the kinetics of grain nucleation and growth. This is essential in thermomechanical processing with grain refinement optimization, aimed at increasing the nucleation rate and reducing the growth rate in order to improve the strength and toughness of finished products, with a notable effect on microstructural evolution.

$$d_s = A d_o^{0.16} \varepsilon^{-0.58} Z^{-0.083} \left[ (exp(513\ \text{kJ/mol}/RT))^{-0.11} \right] \tag{9}$$

The results of physical simulation by analytical and phenomenological methods indicated that the driving force for softening is the energy stored in the strained matrix, which is reduced by SRV and SRX. Since both processes relieve this energy, the progress of one will reduce the driving force available to the other; SRV and SRX are considered concurrent and coupled through stored energy. This microstructural evolution is more complex in the presence of strain-induced precipitates, because the development of their strained state depends on hardening and softening processes, which are strongly influenced by metallurgical factors. Furthermore, the results showed that much of the softening is due to SRV, which makes it the mechanism for controlling and monitoring thermomechanical processing, resulting in grain refinement and strain-induced precipitation.

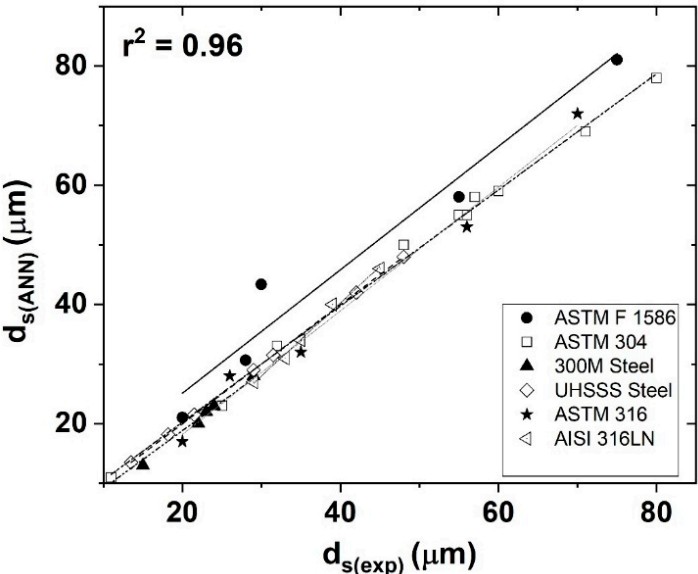

**Figure 17.** Correlation between the experimental recrystallized grain size ($d_{s(exp)}$) and the one calculated via artificial neural network ($d_{s(ANN)}$).

### 3.9. SEM/EDS Microanalysis of Strain-Induced Precipitates

The behavior of the static softening fraction ($X_s$) in response to the time between passes ($t_p$) indicated that the effect of the amount of strain on the interaction between recrystallization and precipitation becomes visible through the formation of the plateau, characterizing the effective action of strain-induced precipitation at the grain boundaries in the initial stages of nucleation (see Figures 16 and 18). As can be seen, precipitation occurs in the condition of 1050 °C, with the presence of a plateau, before static softening reaches 50% at $\varepsilon = 0.30$ in $t_p = 1.0$ s. However, at lower strains, e.g., $\varepsilon = 0.15$, the plateau begins in 10 s with a softening of 30%. Obviously, this precipitation does not occur under the thermodynamic equilibrium inherent in the point of solubility.

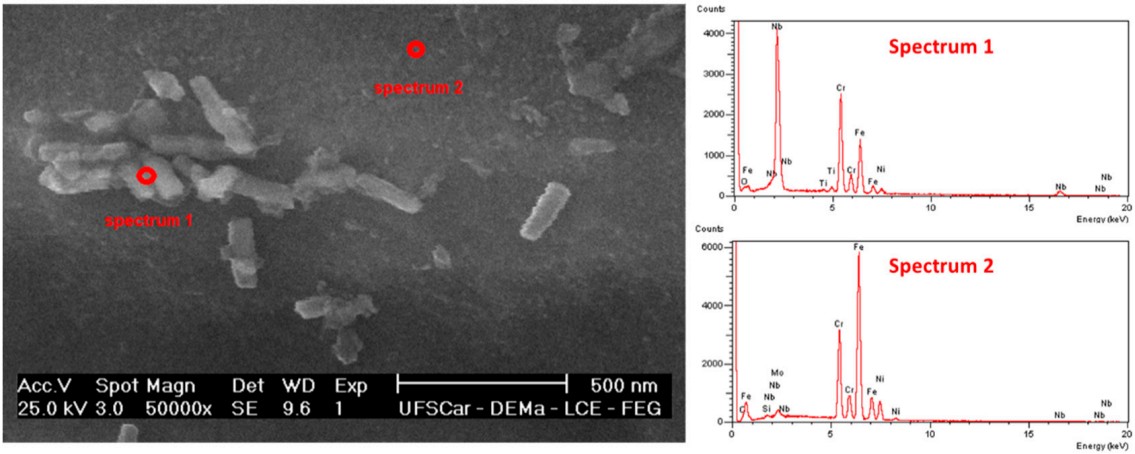

**Figure 18.** SEM/EDX images of fine Z phase precipitates (51.66% Nb, 23.97% Cr, 9.23% N) and of the austenitic matrix (61.32% Fe, 21.16% Cr, 9.88% Ni) in the condition of T = 1100 °C and $t_p = 10$ s.

The formation of Z phase precipitates is attributed to the precipitation of the MDRX phase containing Nb in the transformed austenitic matrix, which can strengthen highly stable steel [51]. A SEM/EDS analysis indicated that Z phase was the main precipitate present in the thermomechanical cycle employed. This phase was composed of 23.97% Cr, 51.66% Nb and 9.23% N and was identified at the grain boundaries with an orientation

relationship of $[010]Z//[110]\gamma$, $[111]Z//[100]\gamma$, $[110]Z//[001]\gamma$, which affected its mobility, as shown in Figure 18. Ornhagen [52] analyzed the composition of Z phase in a similar steel (ISO 5832-9: 1992E) and reported 25% Cr, 61% Nb and 7% N. K.L. Silva et al. [53], who used the precipitate extraction technique, reported that the chemical composition of the Z phase was 32% Cr, 63% Nb and 5% N, which was confirmed by X-ray diffraction. Mariana et al. [54] examined the microstructure of Z phase by TEM/EDX and found a similar composition.

A. Hermant et al. [55], who studied the behavior of grains under hot deformation, stated that the type and amount of microalloying elements (Nb, Cr and N) affect the shape and nature of newly formed precipitates, strengthening the matrix during hot rolling passes, according to the recrystallization mechanisms of AISI 316Nb steel. Moreover, solute drag (Nb-N) can have a significant effect on the kinetics of SRX and precipitation. Therefore, it is important to understand the softening mechanisms in microstructural evolution under hot deformation. It should be noted that during the double-pass thermomechanical processing of ASTM F-1586 steel, the growth rate of SRX grains is strongly affected by the presence, fraction and size of Z phase precipitates that interact with the grain boundaries, hindering their mobility, and that knowledge about these mechanisms is essential for the efficient control of recrystallization kinetics.

## 4. Conclusions

In this work, the static softening of ASTM F-1586 steel was studied by means of double-pass hot compression tests performed in the temperature range of 1000 to 1200 °C and with time between passes ($t_p$) of 1.0 to 1000 s, after reaching strain rates of 0.15 and 0.30 in the 1st pass, thus achieving promising results in optimizing thermomechanical processing by applying the ANN. The following conclusions were reached:

1. Physical simulation of continuous isothermal hot compression tests showed intense DRV, as indicated on the stress–strain curves, followed by a delay in the onset and progress of DRX;

2. The stress–strain curves of the double-pass hot compression tests indicate that ASTM F-1586 steel underwent MDRX at $\varepsilon > 0.30$, and that the softening fraction ($X_s$) increases the higher the temperature and the applied strain;

3. The $X_s$ vs. $t_p$ curves showed high levels of softening ($X_s \sim 40\%$) caused by SRV during short times between passes ($t_p < 1.0$ s), even before the onset of SRX. This was attributed to the moderate stacking fault energy ($\gamma_{sfe} = 69$ mJ/m$^2$), which inhibits the action of thermally activated mechanisms, generating differences in softening fractions ($X_s$) and recrystallization ($X_{srx}$), diverging from the Avrami sigmoidal behavior;

4. The validation of the ANN model in comparison to Arrhenius–Avrami type constitutive models presented satisfactory results and the desired reliability ($R^2 = 0.94$, $\Delta = 0.67\%$) in estimating the softening fraction ($X_s$) and the mean grain size ($d_s$) of ASTM F-1586 steel. However, in the regime characterized by marked strain-induced precipitation, this prediction cannot be adjusted because the model does not include this mechanism;

5. The microstructure subjected to microanalysis (OM/SEM/TEM) showed the presence of elongated grains, partially recrystallized grains and fine Z phase precipitates at the grain boundaries. These grains and precipitates delayed the SRX kinetics, resulting in a low Avrami exponent ($n$), and were responsible for the formation of plateaus in the $X_s$ vs. $t_p$ softening curves above 1050 °C, with the onset ($P_i$) and end ($P_f$) of strain-induced precipitation.

**Author Contributions:** Conceptualization, G.A.d.S.S., S.F.R., C.A.J., F.S. and E.S.S.; methodology, G.A.d.S.S., M.V.G.R., M.N.d.S.L., R.d.C.P.L., D.F.S.d.S., G.M.E.M. and S.F.R.; software, G.A.d.S.S., M.N.d.S.L., D.F.S.d.S., H.F.G.d.A., F.S. and E.S.S.; validation, G.S.R., E.S.S., G.M.E.M., H.F.G.d.A., C.A.J., F.S. and S.F.R.; formal analysis, G.A.d.S.S., M.V.G.R., M.N.d.S.L., R.d.C.P.L., H.F.G.d.A., C.A.J., F.S., S.F.R. and E.S.S.; investigation, G.A.d.S.S., M.V.G.R., E.S.S., G.S.R., F.S., H.F.G.d.A. and S.F.R.; resources, G.S.R., H.F.G.d.A., C.A.J., E.S.S. and S.F.R.; data curation, M.N.d.S.L., R.d.C.P.L., D.F.S.d.S., G.M.E.M. and C.A.J.; writing—original draft preparation, G.A.d.S.S., M.V.G.R., F.S., S.F.R., D.F.S.d.S., G.M.E.M. and E.S.S.; writing—review and editing, E.S.S., G.S.R., H.F.G.d.A., F.S., C.A.J. and S.F.R.; supervision, E.S.S., G.S.R. F.S. and S.F.R.; project administration, E.S.S., G.S.R.. H.F.G.d.A. and S.F.R.; funding acquisition, E.S.S., G.S.R. and S.F.R. All authors have read and agreed to the published version of the manuscript.

**Funding:** This research was funded by the Research and Support Foundation of Maranhão (FAPEMA), grant number 01006/19 and the Brazilian National Council for Scientific and Technological Development (CNPq), grant numbers 403678/2021-8 and 302893/2021-0.

**Data Availability Statement:** The data presented in this study are available in the manuscript itself.

**Acknowledgments:** The authors acknowledge with gratitude funding received from the Brazilian National Council for Scientific and Technological Development (CNPq) and the Research and Support Foundation of Maranhão (FAPEMA). They also thank the team of the Materials Characterization Laboratory (LACAM) and Analytical Central from Federal University of Ceará (UFC) and the PRPGI-IFMA for their unrestricted support. The authors also thank the Laboratory of Structural Characterization (LCE/DEMa/UFSCar/Brazil) for the use of its facilities.

**Conflicts of Interest:** The authors declare no conflict of interest.

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
