# Peer review of "Optimization of Thermomechanical Processing under Double-Pass Hot Compression Tests of a High Nb and N-Bearing Austenitic Stainless-Steel Biomaterial Using Artificial Neural Networks"

_metals, doi:10.3390/met12111783_

Round 1
Reviewer 1 Report
The paper developed an artificial neural network (ANN) to optimize the thermomechanical behavior of stainless-steel biomaterial in a double-pass hot compression test, which consists of calculating the static softening fraction (X s ) and mean recrystallized grain size (d s ), implementing an ANN based on data obtained from hot compression tests. The constitutive analysis and the experimental and ANN-simulated results were in good agreement, indicating that the use of this ANN allows one to optimize the ideal thermomechanical parameters for distribution and refinement of grains with better mechanical properties. The work is interesting and the paper is well represented. So, the reviewer recommends it for publication.
Author Response
Dear reviewer, thanks for your valuable comments
Reviewer 2 Report
This manuscript is well written and acceptable of publication. I recommend it for publication in its present form.
Author Response
Dear reviewer, thanks for your nice words and understanding the presented results. We really appreciate your comments and recommendation.
Reviewer 3 Report
- It is suggested that the process that has been investigated (double-pass hot compression test) in this research should be mentioned in the title.
- The number of references mentioned in the introduction section is small. Please include more references in this section so that readers can have a better understanding of the subject.
- The innovation and difference of this research compared to similar research should be clearly mentioned at the end of the introduction section. At present, this matter has been generalized.
- It is better to use complete words instead of using symbols in section . Materials and Methods. like the: length (l) = 10 mm.
- What is the thickness of molybdenum disk?
- Authors should provide some pictures from experimental tests.
- In Figure 2, what is the reason for the small difference between the stress-strain diagrams at temperatures of 1150 and 1100 degrees Celsius?
- Why did the authors not provide a complete interpretation and explanation of Figure 6?
- How have the authors confirmed the results of the ANN? What is their index?
Author Response
Reviewer 3
- It is suggested that the process that has been investigated (double-pass hot compression test) in this research should be mentioned in the title.
Answer: Dear reviewer, thanks for your great suggestion. We updated the title of the manuscript according to your suggestion.
- The number of references mentioned in the introduction section is small. Please include more references in this section so that readers can have a better understanding of the subject.
Answer: Thank you very much for point this out. We have now included more references in the introduction part.
- The innovation and difference of this research compared to similar research should be clearly mentioned at the end of the introduction section. At present, this matter has been generalized.
Answer: Thanks for suggesting this improvement to our paper. We have included the novelty of this work at the end of the introduction part. The text is now changed as follow:
“The novelty of the developed ANN proposed in this investigation enables one to esti-mate the level of mechanical strength under different conditions and evaluate the hardening and softening mechanisms of this steel by taking into consideration the in-trinsic physical-metallurgy variables of the thermomechanical process”
- It is better to use complete words instead of using symbols in section . Materials and Methods. like the: length (l) = 10 mm.
Answer: Thanks for calling our attention about this point. We have followed your suggestion.
- What is the thickness of molybdenum disk?
Answer: Thanks for your valuable contribution on this. We have included in the text this information.
- Authors should provide some pictures from experimental tests.
Answer: Thanks for your suggestion but the authors judged that the Figures presented in this section is enough for the well understanding of the methodology and avoid the inclusion of many figures in the paper. Your suggestion is being considered for future publications.
- In Figure 2, what is the reason for the small difference between the stress-strain diagrams at temperatures of 1150 and 1100 degrees Celsius?
Answer: Thanks for your question which is of great importance. We have included an explanation in the manuscript regarding this small difference. The reviewer can see in the manuscript a highlighted sentence about this as follow:
“The flow curves at 1100-1150 °C show closer stress levels. This is due to hot defor-mation which favors the dynamic softening processes (dynamic recovery -DRV and dynamic recrystallization - DRX) to occur simultaneously and predominate over the work-hardening (WH). Under these conditions, the intense thermal vibration facili-tates the diffusion of atoms, the mobility and annihilation of dislocations, contributing to the elimination of dislocations leading to the formation of new grains. The curvature of all stress-strain curves…”
- Why did the authors not provide a complete interpretation and explanation of Figure 6?
Answer: Thanks for calling out attention to this point. We have devoted three paragraphs to interpret and explain this figure 6 and can been seen highlighted in the revised version of the manuscript.
- How have the authors confirmed the results of the ANN? What is their index?
Answer: Thanks for your valuable question. We have confirmed the results of the ANN figures 11, 12, 13 and 14 plus description and discussion of each one of them. Figure 14 shows how good were the indexation of each employed ANN represented by the R2.
Reviewer 4 Report
My comments are in a separate file.

Author Response
Reviewer 4
This manuscript demonstrates the application of ANN in predicting the thermomechanical behavior of ASTM F-1586 steel under double-pass hot compression tests. The use of ANN will be important for materials science. I recommend the paper for publication after minor revision.
Answer: Dear reviewer, thanks for your nice words and understanding the presented results. We really appreciate your comments and recommendations.
Line 197 What is DRV?
Answer: Thanks for calling out attention to this point. We have included this information into the paper and highlighted it.
Figure 3, What does dθ/dσ mean?
Answer: Thanks for your nice question. We have included meaning of this relation which is the work hardening rate.
Line 343-345. “the coalescence of precipitates favors grain boundary mobility.” Meaning is not clear.
Answer: Thanks for your good question: Actually, the coalescence mean that the precipitates start to saturate and increase which favor grain mobility leading to more effective softening mechanisms.
Eq. (4), Please cite a paper to understand Eq. (4).
Answer: Thanks for your suggestion. We have included the reference as recommended.
Fig. 15 (a) and (b). The magnification difference of left (OM) and right (SEM) does not seem to be twice. Are they all right?
Answer: Thanks for point this out. Yes, the magnifications and all scale bars are corrected.
Line 604. “tp” --> “tp”
Answer: Thank you very much point this out. We have now corrected accordingly.
Line 607-608. What is non-recrystallization temperature?
Answer: Thanks for your question. Basically, the Tnr is the temperature in which a metal under deformation above this will experiment total recrystallization and below it with only suffer partial recrystallization. This is a well-known point among the metallurgists.
Line 625, Figure17. ds(ANN) is obtained by using eq (9). Is Q=513 kJ/mol applicable to your case? Do you obtain Z by using ANN?
Answer: Thank you very much point this out. Actually, the Q value was obtained before the application of the ANN, and included into the equation. Z is a results of the variables before and and after the application of the ANN.
Line 648. Figure 17 --> Figure 18
Answer: Thank you very much point this out. We have now corrected accordingly
Round 2
Reviewer 3 Report
The manuscript can be accepted in the present form.